# Protocol for development of a reporting guideline (TRIPOD-AI) and risk of bias tool (PROBAST-AI) for diagnostic and prognostic prediction model studies based on artificial intelligence

Gary S Collins [ID],[1,2] Paula Dhiman [ID],[1,2] Constanza L Andaur Navarro [ID],[3] Jie Ma [ID],[1] Lotty Hooft,[3,4] Johannes B Reitsma,[3] Patricia Logullo [ID],[1,2] Andrew L Beam [ID],[5,6] Lily Peng,[7] Ben Van Calster [ID],[8,9,10] Maarten van Smeden [ID],[3] Richard D Riley [ID],[11] Karel GM Moons[3,4]

GSC and KGM contributed equally.

For numbered affiliations see end of article.

**Correspondence to**
Professor Gary S Collins;
gary.collins@csm.ox.ac.uk

## ABSTRACT

**Introduction** The Transparent Reporting of a multivariable prediction model of Individual Prognosis Or Diagnosis (TRIPOD) statement and the Prediction model Risk Of Bias ASsessment Tool (PROBAST) were both published to improve the reporting and critical appraisal of prediction model studies for diagnosis and prognosis. This paper describes the processes and methods that will be used to develop an extension to the TRIPOD statement (TRIPOD-artificial intelligence, AI) and the PROBAST (PROBAST-AI) tool for prediction model studies that applied machine learning techniques.

**Methods and analysis** TRIPOD-AI and PROBAST-AI will be developed following published guidance from the EQUATOR Network, and will comprise five stages. Stage 1 will comprise two systematic reviews (across all medical fields and specifically in oncology) to examine the quality of reporting in published machine-learning-based prediction model studies. In stage 2, we will consult a diverse group of key stakeholders using a Delphi process to identify items to be considered for inclusion in TRIPOD-AI and PROBAST-AI. Stage 3 will be virtual consensus meetings to consolidate and prioritise key items to be included in TRIPOD-AI and PROBAST-AI. Stage 4 will involve developing the TRIPOD-AI checklist and the PROBAST-AI tool, and writing the accompanying explanation and elaboration papers. In the final stage, stage 5, we will disseminate TRIPOD-AI and PROBAST-AI via journals, conferences, blogs, websites (including TRIPOD, PROBAST and EQUATOR Network) and social media. TRIPOD-AI will provide researchers working on prediction model studies based on machine learning with a reporting guideline that can help them report key details that readers need to evaluate the study quality and interpret its findings, potentially reducing research waste. We anticipate PROBAST-AI will help researchers, clinicians, systematic reviewers and policymakers critically appraise the design, conduct and analysis of machine learning based prediction model studies, with a robust standardised tool for bias evaluation.

## Strengths and limitations of this study

► The reporting of clinical prediction models using artificial intelligence is poor.
► There are no guidelines for the reporting or risk of bias assessment of clinical prediction models using artificial intelligence.
► The strengths of this study is that it follows published guidance from the EQUATOR Network for developing reporting guidelines.
► Expert opinion and consensus will be obtained from multiple stakeholders (statisticians, clinician scientists, epidemiologists, computer scientists, funders, healthcare policy-makers, patients and industry leaders).

**Ethics and dissemination** Ethical approval has been granted by the Central University Research Ethics Committee, University of Oxford on 10-December-2020 (R73034/RE001). Findings from this study will be disseminated through peer-review publications.
**PROSPERO registration number** CRD42019140361 and CRD42019161764.

## BACKGROUND

Models that predict clinical outcomes are abundant in the medical literature and are broadly categorised as those that estimate the probability of the presence of a particular outcome (diagnostic) or whether a particular outcome (eg, event) will occur in the future (prognostic).[1] Traditionally, these models (herein referred to as prediction models) have been developed using regression-based methods, typically logistic regression for short-term outcomes and Cox regression for longer-term outcomes.[2] Numerous reviews have observed that studies describing the

development and validation (including updating) of a prediction model often fail to report key information to help readers judge the methods and have a complete, transparent and clear picture of the model's predictive accuracy and other relevant details such as the target population and the content of the model itself.[3–6] The absence of full and comprehensive reporting limits the usability of the findings of these studies, for example, in subsequent validation studies, evidence synthesis studies or in daily practice, and therefore, contribute to research waste.[7] In response to this, in 2015, the Transparent Reporting of a multivariable prediction model for Individual Prognosis Or Diagnosis (TRIPOD) Statement was published.[1 8] The TRIPOD Statement is a checklist of 22 items that authors should report with sufficient detail and clarity to inform how the study was carried out.

Since the publication of the TRIPOD Statement, artificial intelligence (AI) and in particular machine learning, approaches to clinical prediction have evolved and grown in popularity with the number of AI and machine learning publications rapidly rising.[9–14] This is evident within a recent review of COVID-19 related prediction models, where 57 (out of 107 included studies) used machine learning methods to develop their model.[15]

Machine learning, a branch of AI, can be broadly described as data analytical methods that learn from data without being explicitly programmed, with patterns identified based on the data itself. They are often described as having flexibility to capture complex associations particularly in large and unstructured data and complexity in modelling. While the vast majority of the items in the TRIPOD Statement are relevant to machine learning based prediction model studies, there are some unique challenges with machine learning that are not captured. Due to their complexity, these prediction models are typically considered to be 'black box', unlike say regression-based models where the full model can be transparently presented (eg, as an equation containing all the regression coefficients). Also, while many machine learning methods have origins in the statistical literature, two (overlapping) prediction model cultures have emerged as those from a statistical/epidemiological background and those from the computer science/data sciences.[16] Although there is clear overlap, different approaches to model development, validation and updating have appeared, and different and sometimes conflicting terminology have arisen.

Due to the relative novelty of applying machine-learning methods to clinical prediction modelling, there is little information on the quality of reporting of these studies. However, the few reviews that have examined the completeness of reporting of have concluded that reporting is poor.[17 18] In response to these concerns, guidance is required to help authors fully describe their prediction model study when machine learning methods were used. Therefore the TRIPOD group initiated a large international project to develop a consensus based extension of TRIPOD with specific focus on reporting of studies that undertake the development, validation or updating of a diagnostic or prognostic prediction model, using machine learning techniques—herein referred to as TRIPOD-AI.[19] The TRIPOD-AI extension, comprising a checklist and an accompanying elaboration and explanation document will provide researchers, authors, reviewers, editors, users and other stakeholders of machine-learning-based prediction model studies, with guidance on the minimal set of items to report, with detailed examples of good reporting for each item.

Complete reporting allows studies to be understood, replicated and used. However, critical appraisal and of the quality of study method is a crucial component of evidence-based medicine as well. Critical appraisal and assessing the quality of studies is a crucial component of evidence-based medicine. In 2019, the Prediction model Risk Of Bias ASsessment Tool (PROBAST) was published[20 21] to help a variety of stakeholders including, for example, systematic reviewers, researchers, journal editors, manuscript reviewers and policy-makers involved in clinical guideline development, critically appraise the study design, conduct and analysis of prediction model studies. PROBAST comprises four domains (participants, predictors, outcome and analysis) and contains 20 signalling questions to facilitate risk of bias assessment. Clearly risk of bias assessment and reporting are intrinsically linked, in that judging risk of bias is predicated on what has been reported in the primary study. While in principle PROBAST is relevant for prediction model studies using machine learning, different approaches to model development and validation, and terminology have appeared, and the ability to critically appraise these studies is crucial before they are implemented.[22 23] Therefore, in parallel with the development of TRIPOD-AI, we will also develop PROBAST-AI, a tool to assess risk of bias in machine learning based multivariable prediction model studies.

## FOCUS OF TRIPOD-AI AND PROBAST-AI

The focus of both TRIPOD-AI and PROBAST-AI is on reports of research or endeavours in which a multivariable prediction model is being developed (or updated), or validated (tested) using any (supervised) machine learning technique. Conforming to the original TRIPOD and PROBAST publications, a multivariable prediction model is defined as any combination or equation of two or more predictors that is to be used for individualised predictions to estimate an individual's probability of having (diagnosis) or developing (prognosis) a particular health outcome or state. Predictors may have any form and emerge from patient history, physical examination, diagnostic, prognostic or monitoring tests and from undergone treatments. Outcomes may also have any form (dichotomous, categorical, continuous) and of any kind, such as, a particular condition or disorder being present or absent (diagnostic outcome or classification), short-term prognosis outcomes (eg, hospital mortality or postoperative complications), and long-term

prognostic outcomes such as 1-year occurrence of treatment complications, 5-year occurrence of metastases or lifelong survival).

As per the original publications, TRIPOD-AI and PROBAST-AI will also address prediction model studies from all medical care settings (public health, primary, secondary, tertiary and nursing home care) and all corresponding target populations (healthy individuals, suspected and diseased individuals).

TRIPOD-AI and PROBAST-AI are not meant to address:

► Comparative studies that quantify the impact of using a prediction model as compared with not using the model.[24]
► So-called predictor finding studies (also known as risk or prognostic factor studies) where multivariable machine learning techniques are used to identify (usually from a wider set of potential predictors) those predictors that are associated with an outcome, but not to develop a model that can be used for individualised predictions in new individuals.
► Single medical test studies that use machine learning or AI techniques aimed to read, for example, CT or MRI, images to find which image parameters are best associated with an outcome (such studies fall under the remit of STARD-AI[25]). If these image parameters are included as predictors in a multivariable model combined with other predictors, TRIPOD-AI and PROBAST-AI may be useful.

## METHODS/DESIGN
Both TRIPOD-AI and PROBAST-AI will be developed following published guidance from the EQUATOR Network.[26] We will develop the guideline in five stages: (1) systematic reviews to establish the quality of current reporting, (2) Delphi exercise, (3) consensus meeting, (4) development of the guidance statement and (5) guideline dissemination. We have registered our intent to develop the TRIPOD extension for AI on the EQUATOR Network website (www.equator-network.org), the TRIPOD website (www.tripod-statement.org) and recently announced it in the Lancet,[19] while the PROBAST-AI development has been announced on the PROBAST website (www.probast.org).

### TRIPOD-AI/PROBAST-AI working group
The TRIPOD/PROBAST working group will include: (1) an executive committee (2) an advisory and working group and (3) a large international Delphi panel.

The TRIPOD-AI/PROBAST-AI executive committee will be responsible for the leadership and coordination of all the processes involved in the development and dissemination of the TRIPOD-AI guideline. The executive committee consists of the two lead authors of the TRIPOD reporting guideline and the PROBAST tool, and also prediction model experts and researchers from the machine learning community. Key stakeholders for stage 2 (Delphi survey) will be identified and approached to participate and a subset of these key stakeholders (the

advisory group) will participate in stage 3 (consensus meeting).

Here, the term key stakeholder refers to a cross-sector participant (both industry and public sector) who falls into at least one of the following categories:

1. Researchers who have used machine learning in the context of clinical prediction, have clear knowledge and expertise in using machine learning or developed machine learning methods. These include applied (bio)medical investigators, statisticians, epidemiologists and data scientists).
2. Assessors and approvers of AI or machine learning model, such as regulatory assessors and ethics committee members.
3. Beneficiaries or users of the resultant TRIPOD-AI guidance and PROBAST-AI tool such as journal editors and journal reviewers.
4. Commissioners of research grants, such as funders.
5. Consumers of research results such as healthcare providers and patients and citizens.

### Stage 1: systematic review of current reporting
Two parallel systematic reviews are ongoing to evaluate the quality of current reporting in published studies developing, validating or updating machine learning based prediction models in the medical domain. Both systematic reviews will assess adherence of the reporting against the original TRIPOD Statement,[1 8] using the TRIPOD adherence checklist.[27] The reviews will also examine the methodological conduct of the primary studies, including a risk of bias assessment using the recently issued risk of bias tool (quality appraisal) for diagnostic and prognostic prediction model studies (PROBAST),[20 21] and will draw out specific issues, currently not covered by TRIPOD and PROBAST relating to machine learning. The protocols for the two systematic reviews have been registered with the International Prospective Register of Systematic Reviews (PROSPERO IDs CRD42019140361 and CRD42019161764). One review (CRD42019161764) will examine the quality of reporting of machine-learning-based prediction model studies across all medical fields (between January 2018 to December 2019), while the other review (CRD42019140361) will focus on the quality of reporting of machine learning based prediction model studies published in oncology (between January 2019 and September 2019).

Undertaking these reviews serves two purposes: (1) to understand the completeness of current reporting of machine-learning-based prediction model studies in the medical literature and (2) to identify unique reporting items for consideration for TRIPOD extension, and unique risk of bias or quality items for PROBAST extension. The data collection for this phase is underway. The reviews will evaluate the current completeness of reporting and the quality of the research and identify additional reporting and quality items to be considered for TRIPOD-AI and PROBAST-AI.

These two reviews will evaluate the current completeness of reporting and the quality of the research. Together with other evidence[3 4 17 18 28] from existing methodological guidance papers, they will provide important information on the transparency and quality of reporting. Using the original TRIPOD and PROBAST checklists as starting points, the executive committee will identify in the literature the preliminary items to consider in stage 2 (the Delphi study) and therefore inclusion in the eventual TRIPOD-AI checklist and PROBAST-AI tool.

### Stage 2: Delphi exercise

We will perform an extensive Delphi survey among a large international network of relevant stakeholders, with a maximum of three rounds, to help decide on items that could be modified, added to, or removed from the TRIPOD 2015 checklist to form the TRIPOD-AI checklist, and subsequently the PROBAST-AI checklist.

### Design

The Delphi process will comprise of a series of rounds where panellists will independently and anonymously evaluate and achieve consensus on the inclusion or exclusion of the proposed reporting and quality items—in addition to suggesting additional items. The process will be repeated for a maximum of three rounds. Following each round, participants will be provided with structured feedback of the previous round to help reconcile individual opinions and achieve group consensus. Items achieving a high level of agreement (≥70%) will be taken forward to the consensus meeting (stage 3).

### Selection of potential items

The list of items for TRIPOD-AI (and PROBAST-AI) will be collated by the executive committee, including the results of the two systematic reviews, any other available studies on methodology or reporting of machine learning based prediction models, and expert recommendations from the Delphi panellists. Relevant methodological guidance or methodological papers will be retrieved to identify additional candidate reporting and quality items for machine-learning-based prediction model studies. Preselection involves dividing items into those to further consider, those that can be provided as optional guidance (to be outlined in an Explanation and Elaboration accompanying document), or those not to consider for potential inclusion. Delphi participants will have the opportunity to view and provide feedback in each round, and also to suggest new items.

### Recruitment process and participants

Delphi participants will be identified through professional networks of the executive committee, participation in the Delphi exercise of the original TRIPOD guideline (and TRIPOD for Abstracts and TRIPOD Cluster Delphi surveys), original PROBAST Delphi exercise, via self-response to the Lancet 2019 paper where TRIPOD-AI was announced,[19] and responses to social media announcements of TRIPOD-AI (eg, Twitter).

We will invite international participants with diverse roles (eg, researchers, healthcare professionals, journal editors, funders, policy makers, healthcare regulators, end users of prediction models) from a range of settings (eg, universities, hospitals, primary care, biomedical journals, non-profit organisations and for-profit organisations). Participants will be invited via personalised email that will describe the TRIPOD-AI extension and PROBAST-AI tool development, and explain the objective, process, and timelines of the Delphi exercise. We plan to invite at least 200 participants to the Delphi survey. In all rounds, the survey will remain open for 3 weeks, with a reminder email sent 1 week after the initial invitation. In round two of the Delphi exercise, additional participants may be sought to ensure fair representation of all key stakeholders.[29]

Informed consent from participants will be obtained using an online consent form and participants can withdraw at any time. Individuals who indicate that they wish to opt out of the survey will be removed from subsequent invitations. Participants will not know the identities of other individuals in the Delphi panel, nor will they know the specific answers that any individual provides.

### Procedure for selection of items

We plan to ask participants to consider the following guiding principles when reviewing existing, new or modified items for inclusion: (1) reporting of the item should facilitate reproducibility of the study (ie, users should be able to recreate the findings based on the information reported); (2) reporting of the item facilitates assessment of the quality and risk of bias in and applicability of the machine learning study findings, to enhance their uptake and use in subsequent studies, systematic reviews and daily practice; (3) item is likely relevant to nearly all prediction model studies; (4) the set of items represent the minimum that should be reported in all machine learning studies developing, validating or updating a diagnostic or prognostic prediction model.

### Round 1

Participants will be asked to rate on a 5-point Likert scale, the extent to which they agree with the inclusion of each checklist item in the TRIPOD-AI extension and PROBAST-AI tool (1=strongly disagree, 2=somewhat disagree, 3=I don't know, 4=somewhat agree, 5=strongly agree). A free-text box will be provided for general comments on each item (to justify their decision or suggest wording changes), and a free-text box will be provided at the end of the survey to suggest additional checklist items or provide general comments on the checklist. The survey will be pilot-tested for usability and clarity to a small number of individuals familiar with prediction models or machine learning but not involved in the TRIPOD-AI guideline extension or PROBAST-AI tool and revised accordingly based on their feedback.

## Round 2

The same participants involved in round 1 will be invited to participate in round 2. Participants will be provided with their first-round responses on each item, an anonymised summary of the group ratings and anonymised comments to justify ratings. Using the same format as round 1, participants will be presented with each item, including any new items suggested during round 1, and again express the extent to which they agree with the inclusion of the item in the TRIPOD-AI checklist or PROBAST-AI tool, considering the structured feedback to inform their responses. Participants who were invited to participate in round 1, but who did not respond will be invited to participate in round 2, and will be presented with an anonymised summary of the group ratings. Items that reached a high-level of agreement (scoring 4 or 5) in round 1 (≥70%) will be presented for information purposes only, with no voting on these items, though a free-text box will be provided for any comments. A third Delphi round will be used if deemed necessary by the Executive Committee.

### Results from the Delphi survey

Item scores will be summarised for the entire panel as a whole, as appropriate (eg, frequency and proportions across the rating categories) accompanied by a narrative summary of findings, comments, and suggestions. Results from both rounds of the survey will be discussed by the executive committee. For items where there was no consensus following the second Delphi found will be discussed by the executive committee, and will be considered for discussion at the subsequent consensus meeting.

### Stage 3: consensus meeting

Two virtual consensus meetings (separately for TRIPOD-AI and PROBAST-AI), both spread over 2 days, will be held with the objective of discussing the results from the Delphi exercise and finalising items to be included in the reporting guideline and risk of bias tool. The composition of the consensus group will reflect the diversity of the key stakeholders addressed above. Key experts participating in the Delphi exercise will be considered to participate in the consensus meeting. We will also consider inviting experts who did not contribute to the Delphi to participate in the consensus. A total of around 25–30 international participants are expected to contribute to the virtual consensus meeting.

### Procedure

The agenda and any material (eg, results from the systematic reviews and Delphi) for the consensus meeting will be prepared by the executive committee and will be shared with attendees in advance. Members of the executive committee will facilitate a structured discussion on the rationale behind each item identified in the Delphi exercise. Consensus meeting participants will then be given the opportunity to discuss each item (reporting item for TRIPOD-AI and signalling question for PROBAST-AI),

and vote on each item. The decision to retain an item in the TRIPOD-AI and PROBAST-AI will be based on achieving at least 70% support from the consensus meeting participants. The group will agree on the draft list of reporting items for the final TRIPOD-AI extension and PROBAST-AI tool. Specific item wording will not be discussed during the meeting, though participants can suggest and the group to agree on general intent and meaning of the item. Plans for dissemination will be discussed at the end of the consensus meeting.

### Pilot testing

We will invite authors of machine learning prediction model studies in the medical domain, doctoral students undertaking prediction model, machine learning courses or workshops, and peer-reviewers and editors of journals who frequently publish such prediction model studies, to pilot the use of a draft version of the TRIPOD-AI checklist and PROBAST-AI tool. We will ask those who pilot the checklist and tool whether the wording of items is ambiguous or difficult to interpret.

### Stage 4: development of the draft TRIPOD-AI statement, PROBAST-AI and explanation and elaboration documents

The executive committee will lead the development of the TRIPOD-AI reporting guidance and PROBAST-AI signalling questions based on the agreed list of items from the consensus meeting (stage 3). The executive committee will invite a subset of members from the consensus meeting (to form a writing group) to help draft the explanation and elaboration paper.

The executive committee will reserve the right to update (ie, remove or add) additional items to the TRIPOD-AI checklist during the development of the TRIPOD-AI statement, if and as necessary (as a result of the pilot testing).

For each of the TRIPOD-AI extension and the PROBAST-AI risk of bias tool, two manuscripts will be developed: (1) the statement paper, presenting the checklist/tool and describing the process of how it was developed and (2) an explanation and elaboration paper. The explanation and elaboration papers will outline the rationale of the reporting items (TRIPOD-AI) and signalling questions (PROBAST-AI), examples of good reporting (TRIPOD-AI) and examples of how to use PROBAST-AI. Drafts of the papers will be circulated to all participants of the consensus meeting for their comments.

### Stage 5: guideline dissemination

The dissemination strategy will be informed by discussions at the consensus meeting. We will aim to seek simultaneous publication in key journals to target different readerships. To increase visibility and aid uptake, the TRIPOD-AI checklist and PROBAST-AI tool will be published open access, and made available on the TRIPOD website along with other TRIPOD extensions (www.tripod-statement.org), and on the PROBAST website (www.probast.org) respectively, as well on the PROGRESS website (www.

prognosisresearch.com). The TRIPOD-AI extension will be indexed on the EQUATOR website (www.equator-network.org). Social media will be used to help disseminate the extension. The Executive Committee will (and consensus participants will be encouraged to) publicise the TRIPOD-AI statement and PROBAST-AI tool at key conferences and courses.

## PUBLICATION PLAN

It is envisaged that the following publications will arise from the TRIPOD-AI and PROBAST-AI initiative:
► Publication 1: study protocol.
► Publication 2: systematic review protocol (with registration on PROSPERO).
► Publication 3 and 4: Systematic reviews.
► Publication 5 & 6: TRIPOD-AI statement and the Explanation and Elaboration paper.
► Publication 7 & 8: PROBAST-AI tool and the Explanation and Elaboration paper.

## CONCLUSION

The number of prediction model studies using machine learning methods is rapidly increasing, including developed, validated or updated prediction models. Ensuring that key details are reported is important so that readers can evaluate the study quality, and interpret its findings including the developed, validated or updated prediction model to enhance their uptake in subsequent research (eg, validation studies), evidence synthesis projects (eg, systematic reviews of prediction models) and in daily practice by healthcare professionals, patients or citizens. We anticipate that TRIPOD-AI will help authors transparently report their study and help reviewers, editors, policy-makers and end-users understand the methods and findings, and thereby reduce research waste. Similarly, we anticipate PROBAST-AI will help researchers, clinicians, systematic reviewers and policy-makers critically appraise the design, conduct and analysis of machine-learning-based prediction model studies.

**Author affiliations**
¹Centre for Statistics in Medicine, Nuffield Department of Orthopaedics, Rheumatology & Musculoskeletal Sciences, University of Oxford, Oxford, UK
²NIHR Oxford Biomedical Research Centre, John Radcliffe Hospital, NIHR Oxford Biomedical Research Centre, Oxford, UK
³Julius Center for Health Sciences and Primary Care, Utrecht, Utrecht, Netherlands
⁴Cochrane Netherlands, Julius Center for Health Sciences and Primary Care, Utrecht, Utrecht, Netherlands
⁵Department of Epidemiology, Harvard T.H. Chan School of Public Health, Boston, Massachusetts, USA
⁶Department of Biomedical Informatics, Harvard Medical School, Boston, Massachusetts, USA
⁷Google Health, Google, Palo Alto, California, USA
⁸Department of Development and Regeneration, KU Leuven, Leuven, Belgium
⁹Department of Biomedical Data Sciences, Leiden University Medical Centre, Leiden, Netherlands
¹⁰EPI-Centre, KU Leuven, Leuven, Belgium
¹¹Centre for Prognosis Research, School of Medicine, Keele University, Keele, UK

**Correction notice** Since this article was first published online, the author name Ji has been updated to Jie.

**Contributors** GSC, PD, CLAN, JM, LH, JBR, PL, ALB, LP, BVC, MvS, RDR and KGMM were involved in the planning and design of the study. GC drafted the manuscript with all authors contributing to the writing.

**Funding** This research was supported by Health Data Research UK, an initiative funded by UKResearch and Innovation, Department of Health and Social Care (England) and the devolved administrations, and leading medical research charities, Cancer Research UK programme grant (C49297/A27294), the NIHR Biomedical Research Centre, Oxford, and the Netherlands Organisation for Scientific Research.

**Competing interests** None declared.

**Patient and public involvement** Patients and/or the public were not involved in the design, or conduct, or reporting, or dissemination plans of this research.

**Patient consent for publication** Not required.

**Provenance and peer review** Not commissioned; externally peer reviewed.

**ORCID iDs**
Gary S Collins http://orcid.org/0000-0002-2772-2316
Paula Dhiman http://orcid.org/0000-0002-0989-0623
Constanza L Andaur Navarro http://orcid.org/0000-0002-7745-2887
Jie Ma http://orcid.org/0000-0002-3900-1903
Patricia Logullo http://orcid.org/0000-0001-8708-7003
Andrew L Beam http://orcid.org/0000-0002-6657-2787
Ben Van Calster http://orcid.org/0000-0003-1613-7450
Maarten van Smeden http://orcid.org/0000-0002-5529-1541
Richard D Riley http://orcid.org/0000-0001-8699-0735

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
