## [Reviewer comments · BMJ Open]

ARTICLE DETAILS

TITLE (PROVISIONAL)	A protocol for development of a reporting guideline (TRIPOD-AI) and risk of bias tool (PROBAST-AI) for diagnostic and prognostic prediction studies based on artificial intelligence
AUTHORS	Collins, Gary; Dhiman, Paula; Andaur Navarro, Constanza L.; Ma, Ji; Hooft, Lotty; Reitsma, Johannes; Logullo, Patricia; Beam, Andrew; Peng, Lily; Van Calster, Ben; van Smeden, Maarten; Riley, Richard; Moons, Karel

VERSION 1 – REVIEW

REVIEWER	Challen, Robert University of Exeter, EPSRC Centre for Predictive Modelling in Healthcare
REVIEW RETURNED	12-Feb-2021

GENERAL COMMENTS	TITLE: A protocol for development of a reporting guideline (TRIPOD-AI) and risk of bias tool (PROBAST-AI) for diagnostic and prognostic prediction studies based on artificial intelligence SUMMARY: The article defines the need for a specific guideline for reporting AI based diagnostic and prognostic tools in the scientific literature. The article defines the process that development of this guideline will take, and largely follows an established standard Delphi-based process. The plan is clear and There are two parts to this problem that the paper could clarify, both of which relate to scope of the application of the checklist: Firstly, the paper talks about AI but in reality most of it is seems about supervised machine learning classification based predictive and prognostic models, where the training occurs on a “gold standard” dataset. The implication in the “Focus” section is that the scope of this guideline will be largely limited to these techniques, and not I think, designed to include more unsupervised, continuously learning or reinforcement learning AI systems, or systems in which the behaviour of the system is predicting optimal strategies. These come with a large number of additional complexity and safety issues that I think are out of the scope, but it would be good to clarify that. Secondly, it is not completely clear in the paper about the maturity of the studies this guideline is aimed at. I imagine that TRIPOD-AI and PROBAST-AI are designed for studies of pre-clinical research, and the reporting of lab based ML systems, rather than studies of clinical application of ML. Given the complexity of implementation of decision support tools into clinical environments and potentials for complex cognitive biases, I would expect the
---

	guidelines of clinical implementation of AI systems to recommend a randomised clinical trial, which I think is also outside of the scope of this piece of work. Beyond those scope questions I think it is a valuable piece of work and support its publication.
--	--

REVIEWER	Wong, David The University of Manchester, Centre for Health Informatics
REVIEW RETURNED	02-Mar-2021

GENERAL COMMENTS	Thank you for the opportunity to review this protocol. Given the authors' previous experience in developing reporting guidelines and bias tools, the methods presented here are clear and follow a well-tested path. As such, my comments are focused on possibly ambiguities in the text 1.) The 'participants' subsection is an exact replica of portions of the preceding subsection 2.) For Stage 1, I would be keen to clarify whether the protocol accurately reflects current progress. I note from prospero that data collection for one of the studies was intended to be completed at the start of 2020. 3.) The final point of the article summary mentions the potential limitation of the Delphi process, but as far as I can see, this is not mentioned in the protocol itself. If this is indeed the case, then perhaps the easiest thing to do is to remove it from the summary (my understanding is that the strength + limitations is a shoe-horned structure for 'normal' articles that doesn't really fit protocols so well)
--

REVIEWER	Lim, Gilbert National University of Singapore, School of Computing
REVIEW RETURNED	05-Mar-2021

GENERAL COMMENTS	I appreciate the opportunity to review the protocol for this proposed extension to the TRIPOD statement and PROBAST tool, towards improving their application towards machine learning (ML) and artificial intelligence (AI) approaches. This proposal fulfils and urgent need, and the five-stage development methodology is generally clearly specified and justified, and follows the established EQUATOR guideline development framework. A few minor comments/suggestions might however be considered: 1. While the recruitment process for the Delphi exercise (Stage 2) has been stated to take place over multiple channels, there remains no assurance that the final group of selected participants is entirely inclusive and represents all relevant stakeholders comprehensively. Moreover, the stringent multi-round participatory demands might disincentivize certain would-be contributors from joining the Delphi panel (see "A critical review of the Delphi technique as a research methodology for nursing", Keeney et al.; International Journal of Nursing Studies 38 [2001]: 195-200). As such, it might be considered to also publicize an open call/provide an avenue for (possibly-anonymous) suggestions on the protocol to be collected from interested parties, without requiring their full commitment to joining the Delphi panel. These suggestions might then be considered alongside the preliminary
---

	items obtained from the systematic reviews, to possibly be championed by the panelists. 2. While computer scientists are mentioned as a stakeholder group in the article summary, the stakeholder list in the TRIPOD-AI/PROBAST-AI working group section appears tilted towards addressing the biomedical research community. However, given the apparent increased prevalence of applicative publications in general AI/ML venues, with some of the relevant predictive work arising from AI/ML researchers seeking an application for their methods (rather than biomedical researchers starting from a problem to solve), it might be considered to cater some outreach specifically towards the AI/ML communities as well. 3. The Background section mentions that "Also, whilst many machine learning methods have origins in the statistical literature, two (overlapping) prediction model cultures have emerged..."; the relevant paper "Statistical Modeling: The Two Cultures", Brieman; Statistical science 16.3 [2001]: 199-231 might be referenced.
--	---

REVIEWER	Baca-Garcia, Enrique University Hospital Jimenez Diaz Foundation, Psychiatry
REVIEW RETURNED	08-Mar-2021

GENERAL COMMENTS	I WELCOME YOU FOR THIS INITIATIVE. The use of machine learning, without being novel, is growing exponentially in biomedicine (Oquendo, M., A. et al. Machine learning and data mining: strategies for hypothesis generation. Mol Psychiatry 17, 956–959 (2012). https://doi.org/10.1038/mp.2011.173). It is very necessary a guide for authors, reviewers and editors so that there is no "overfitting" in the field. I would like to make two minor contributions to your protocol. You should invite specialists in databases and registries that are the source for machine learning analysis. If the quality of these is not minimally assured the result of any analysis can be a lot of noise. As the authors point out, as machine learning models are black box models they can generate undetectable ambiguities and confusions. I would also suggest incorporate consultants from the software industry. Many of these algorithms are opaque and researchers often do not have full access to their fundamentals. A minor aspect is to include a section to assess the replication of the results with independent samples. It should go beyond the approach of validating the analyses by splitting the sample, with a validation set, "leave one out"...
--

REVIEWER	Nagaratnam, Kiruba Royal Berkshire NHS Foundation Trust, Acute Stroke
REVIEW RETURNED	14-Mar-2021

GENERAL COMMENTS	With the increasing use of AI in healthcare and multiplying number of vendors in the market, it is time we have a system for evaluating and appraising AI algorithm-based predictive models. The authors have clearly described the methodology in the protocol for expanding the TRIPOD and PROBAST tools for this purpose. My only recommendation would be for the authors to consider not only the diversity in roles but also in ethnicity and gender among the Delphi participants. In addition to note a few typos
--

	1. Page 3, line 14 – The strength of study is that it follows.. 2. Page 3, line 15 – Expert opinion and consensus would be obtained 3. Page 5, paragraph 3, line 32 – there appears the sentence repeats. I believe this work will add to our knowledge about evaluating AI predictive models and will directly impact commissioning/procurement decisions and clinical application of similar models in the future. Hence I would strongly recommend it to be published in the BMJ Open.
--	---

REVIEWER	Roper, Marc University of Strathclyde
REVIEW RETURNED	15-Mar-2021

GENERAL COMMENTS	The proposed study addresses a significant and important problem and the protocol presented is well- designed and complete. I just have one question in relation to the Delphi study: in the recruitment process it states that the intention is to invite at least 100 participants to the survey but is there a lower bound on the number of acceptances or a required coverage of expertise necessary for the study to proceed? It would be helpful if this point could be clarified in the protocol.
--

VERSION 1 – AUTHOR RESPONSE

REVIEWERS COMMENTS

Reviewer: 1

Dr. Robert Challen, University of Exeter, Taunton and Somerset NHS Foundation Trust

Comments to the Author:

TITLE: A protocol for development of a reporting guideline (TRIPOD-AI) and risk of bias tool (PROBAST-AI) for diagnostic and prognostic prediction studies based on artificial intelligence

SUMMARY: The article defines the need for a specific guideline for reporting AI based diagnostic and prognostic tools in the scientific literature. The article defines the process that development of this guideline will take, and largely follows an established standard Delphi-based process. The plan is clear.

There are two parts to this problem that the paper could clarify, both of which relate to scope of the application of the checklist:

COMMENT 1: Firstly, the paper talks about AI but in reality most of it is seems about supervised machine learning classification based predictive and prognostic models, where the training occurs on a “gold standard” dataset. The implication in the “Focus” section is that the scope of this guideline will be largely limited to these techniques, and not I think, designed to include more unsupervised, continuously learning or reinforcement learning AI systems, or systems in

which the behaviour of the system is predicting optimal strategies. These come with a large number of additional complexity and safety issues that I think are out of the scope, but it would be good to clarify that.

RESPONSE 1: The reviewer is correct, the focus is on supervised methods. We have amended this under 'Focus of TRIPOD-AI and PROBAST-AI', so that it now reads

'The focus of both TRIPOD-AI and PROBAST-AI is on reports of research or endeavours in which a multivariable prediction model is being developed (or updated), or validated (tested) using any (supervised) machine learning technique.'

COMMENT 2: Secondly, it is not completely clear in the paper about the maturity of the studies this guideline is aimed at. I imagine that TRIPOD-AI and PROBAST-AI are designed for studies of pre-clinical research, and the reporting of lab based ML systems, rather than studies of clinical application of ML. Given the complexity of implementation of decision support tools into clinical environments and potentials for complex cognitive biases, I would expect the guidelines of clinical implementation of AI systems to recommend a randomised clinical trial, which I think is also outside of the scope of this piece of work.

RESPONSE 2: The reviewer is correct, both TRIPOD-AI and PROBAST-AI are in essence pre-clinical studies where ML models are being developed and evaluated for their statistical performance. Subsequent evaluation, say in an randomised controlled trial, to examine their efficacy on patient outcomes, cost-effectiveness, decision-making etc are beyond the scope. However, recent guidance extending the CONSORT statement for evaluating AI interventions (CONSORT-AI) have recently been published (Liu et al, Nature Medicine 2020). There is also similar guidance that bridges the development-to-implementation gap in clinical ML (Vasey et al, Nature Medicine 2021).

Beyond those scope questions I think it is a valuable piece of work and support its publication.

Reviewer: 2

Dr. David Wong, The University of Manchester

Comments to the Author:

Thank you for the opportunity to review this protocol. Given the authors' previous experience in developing reporting guidelines and bias tools, the methods presented here are clear and follow a well-tested path. As such, my comments are focused on possibly ambiguities in the text

COMMENT 3. The 'participants' subsection is an exact replica of portions of the preceding subsection

RESPONSE 3: We thank the reviewer for spotting this and have removed the duplicate text.

COMMENT 4: For Stage 1, I would be keen to clarify whether the protocol accurately reflects current progress. I note from prospero that data collection for one of the studies was intended to be completed at the start of 2020.

RESPONSE 4: We thank the reviewer for the comment. Extraction for both reviews are now complete and have been submitted for publication in February/March 2021.

COMMENT 5: The final point of the article summary mentions the potential limitation of the Delphi process, but as far as I can see, this is not mentioned in the protocol itself. If this is indeed the case, then perhaps the easiest thing to do is to remove it from the summary (my understanding is that the strength + limitations is a shoe-horned structure for 'normal' articles that doesn't really fit protocols so well)

RESPONSE 5: The reviewer is correct, we are following journal formatting requirements.

Reviewer: 3

Dr. Gilbert Lim, National University of Singapore

Comments to the Author:

I appreciate the opportunity to review the protocol for this proposed extension to the TRIPOD statement and PROBAST tool, towards improving their application towards machine learning (ML) and artificial intelligence (AI) approaches. This proposal fulfils and urgent need, and the five-stage development methodology is generally clearly specified and justified, and follows the established EQUATOR guideline development framework. A few minor comments/suggestions might however be considered:

COMMENT 6: While the recruitment process for the Delphi exercise (Stage 2) has been stated to take place over multiple channels, there remains no assurance that the final group of selected participants is entirely inclusive and represents all relevant stakeholders comprehensively. Moreover, the stringent multi-round participatory demands might disincentivize certain would-be contributors from joining the Delphi panel (see "A critical review of the Delphi technique as a research methodology for nursing", Keeney et al.; International Journal of Nursing Studies 38 [2001]: 195-200).

As such, it might be considered to also publicize an open call/provide an avenue for (possibly-anonymous) suggestions on the protocol to be collected from interested parties, without requiring their full commitment to joining the Delphi panel. These suggestions might then be considered alongside the preliminary items obtained from the systematic reviews, to possibly be championed by the panelists.

RESPONSE 6: We thank the reviewer for this comment. Our experience with developing other reporting guidelines (e.g., TRIPOD, GATHER, PROBAST, AGreMA) using a similar process (and a process

following recommended guidelines for achieving consensus for reporting guideline development, e.g., Moher et al PLoS Med 2010) is we don't typically observe fatigue amongst Delphi participants. Furthermore, we plan a maximum 3 Delphi rounds, but in reality and experience from other guideline development initiatives, 2 rounds usually suffices.

With regards to representativeness of the Delphi participants, we have increased the number of invited participants to over 200 to cover a range of relevant stakeholders. After round 1 of the Delphi survey, if we observe lack of response from certain key stakeholders, additional members will be identified for round 2 (e.g., Boel et al J Clin Epidemiol 2021). We have amended the protocol to reflect this.

COMMENT 7: While computer scientists are mentioned as a stakeholder group in the article summary, the stakeholder list in the TRIPOD-AI/PROBAST-AI working group section appears tilted towards addressing the biomedical research community. However, given the apparent increased prevalence of applicative publications in general AI/ML venues, with some of the relevant predictive work arising from AI/ML researchers seeking an application for their methods (rather than biomedical researchers starting from a problem to solve), it might be considered to cater some outreach specifically towards the AI/ML communities as well.

RESPONSE 7: We thank the reviewer for the comment, however our scope and focus is solely on the biomedical research community, we do not attempt to go beyond this.

COMMENT 8: The Background section mentions that "Also, whilst many machine learning methods have origins in the statistical literature, two (overlapping) prediction model cultures have emerged..."; the relevant paper "Statistical Modeling: The Two Cultures", Brieman; Statistical science 16.3 [2001]: 199-231 might be referenced.

RESPONSE 8: We have added this reference.

Reviewer: 4

Prof. Enrique Baca-Garcia, University Hospital Jimenez Diaz Foundation, Autonomous
University of Madrid
Comments to the Author:

I WELCOME YOU FOR THIS INITIATIVE. The use of machine learning, without being novel, is growing exponentially in biomedicine (Oquendo, M., A. et al. Machine learning and data mining: strategies for hypothesis generation. Mol Psychiatry 17, 956–959 (2012). <https://doi.org/10.1038/mp.2011.173>). It is very necessary a guide for authors, reviewers and editors so that there is no "overfitting" in the field.

COMMENT 9: I would like to make two minor contributions to your protocol. You should invite specialists in databases and registries that are the source for machine learning analysis. If the quality of these is not minimally assured the result of any analysis can be a lot of noise. As the authors point out, as machine learning models are black box models they can generate undetectable ambiguities and confusions.

RESPONSE 9: We thank the reviewer for the comment whilst not explicitly mentioned in the protocol (as there are many stakeholders), our invited list of participants cover these specialties.

COMMENT 10: I would also suggest incorporate consultants from the software industry. Many of these algorithms are opaque and researchers often do not have full access to their fundamentals.

A minor aspect is to include a section to assess the replication of the results with independent samples. It should go beyond the approach of validating the analyses by splitting the sample, with a validation set, "leave one out" ...

RESPONSE 10: We thank the reviewer for the comment whilst not explicitly mentioned in the protocol (as there are many stakeholders), our invited list of participants cover these specialties. With regards to the issue on replication this will be determined through the Delphi process.

Reviewer: 5

Dr. Kiruba Nagaratnam, Royal Berkshire NHS Foundation Trust

Comments to the Author:

With the increasing use of AI in healthcare and multiplying number of vendors in the market, it is time we have a system for evaluating and appraising AI algorithm-based predictive models. The authors have clearly described the methodology in the protocol for expanding the TRIPOD and PROBAST tools for this purpose.

COMMENT 11: My only recommendation would be for the authors to consider not only the diversity in roles but also in ethnicity and gender among the Delphi participants.

RESPONSE 11: We fully agree with the reviewer and our list of invited Delphi participants has been created to ensure fair representation of these (and other) diverse groups.

In addition to note a few typos

COMMENT 12: Page 3, line 14 – The strength of study is that it follows..

RESPONSE 12: We thank the reviewer for spotting this and have amended as suggested.

COMMENT 13: Page 3, line 15 – Expert opinion and consensus would be obtained

RESPONSE 13: We thank the reviewer for spotting this and have amended accordingly.

COMMENT 14: Page 5, paragraph 3, line 32 – there appears the sentence repeats.

RESPONSE 14: We have checked the text and unable to see duplicate sentences.

I believe this work will add to our knowledge about evaluating AI predictive models and will directly impact commissioning/procurement decisions and clinical application of similar models in the future. Hence I would strongly recommend it to be published in the BMJ Open.

Reviewer: 6

Dr. Marc Roper, University of Strathclyde

Comments to the Author:

COMMENT 15: The proposed study addresses a significant and important problem and the protocol presented is well- designed and complete. I just have one question in relation to the Delphi study: in the recruitment process it states that the intention is to invite at least 100 participants to the survey but is there a lower bound on the number of acceptances or a required coverage of expertise necessary for the study to proceed? It would be helpful if this point could be clarified in the protocol.

RESPONSE 15: With regards to representativeness of the Delphi participants, we have increased the number of invited participants to over 200 to cover a range of relevant stakeholders. After round 1 of the Delphi survey, if we observe lack of response from certain key stakeholders, additional members will be identified for round 2 (e.g., Boel et al J Clin Epidemiol 2021). We have amended the protocol to reflect this.

VERSION 2 – REVIEW

REVIEWER	Challen, Robert University of Exeter, EPSRC Centre for Predictive Modelling in Healthcare
REVIEW RETURNED	20-Apr-2021

GENERAL COMMENTS	The authors have addressed my concerns. I am sure the editors will go through this with a fine tooth comb but I spotted a bit of word salad that has crept into the abstract: P3 L15: "This paper describes the processes and methods that will be used to develop an extension to the TRIPOD statement (TRIPOD-AI) and the PROBAST (PROBAST-AI) tool for prediction model studies that applied machine learning techniques."
---

REVIEWER	Lim, Gilbert National University of Singapore, School of Computing
REVIEW RETURNED	27-Apr-2021

GENERAL COMMENTS	We thank the authors for addressing our concerns from the previous review round, and have no further comments.
--

REVIEWER	Baca-Garcia, Enrique University Hospital Jimenez Diaz Foundation, Psychiatry
REVIEW RETURNED	01-May-2021

GENERAL COMMENTS	CONGRATULATIONS!!!
--------------------